# Psychometric Properties of the Italian Version of the 25-Item Hikikomori Questionnaire

**DOI:** 10.3390/ijerph192013552

**Published:** 2022-10-19

**Authors:** Simone Amendola, Fabio Presaghi, Alan R. Teo, Rita Cerutti

**Affiliations:** 1Department of Dynamic and Clinical Psychology and Health Studies, Sapienza University of Rome, 00185 Rome, Italy; 2Department of Psychology of Development and Socialization Processes, Sapienza University of Rome, 00185 Rome, Italy; 3Center to Improve Veteran Involvement in Care (CIVIC), VA Portland Health Care System, Portland, OR 97239, USA; 4Department of Psychiatry, Oregon Health & Science University, Portland, OR 97239, USA

**Keywords:** mental health, psychiatry, social detachment, social isolation, social avoidance

## Abstract

A serious form of social withdrawal, initially described within Japan as hikikomori, has received increasing attention from the international scientific community during the last decade. The 25-item Hikikomori Questionnaire (HQ-25) was initially developed and validated in Japan. To date, data on its psychometric properties in other populations where cases of hikikomori have been described are still scarce. Thus, the aims of this study were to (1) translate, adapt, and validate the Italian version of the HQ-25 analyzing its psychometric properties; and (2) verify the association between hikikomori and personality functioning, social support, and problematic Internet use. A sample of 372 Italian adults aged 18 to 50 years completed the HQ-25 and measures of psychoticism, personality dysfunction, social support, and problematic Internet use were employed to test the convergent validity of the HQ-25. The data showed a satisfactory fit for a three-factor model, significantly better than a one-factor model. The three factors (socialization, isolation, and emotional support, as in the original study on the HQ-25) correlated positively with psychoticism, personality dysfunction, and problematic Internet use, and correlated negatively with social support. A lifetime history of hikikomori was present in 1.1% of the sample (*n* = 4). This is the first study to use the Italian validated version of the HQ-25 with an adult population. The findings from this study provide evidence of the satisfactory psychometric properties of the Italian version of the HQ-25 and support further investigation of the HQ-25 as an instrument to help screen for and investigate the presence of hikikomori.

## 1. Introduction

*Hikikomori* indicates to a serious condition of social withdrawal that is a matter of concern and alarm, particularly for urbanized and technologically advanced societies over the last decade [1,2,3]. Hikikomori derives from the Japanese verbs *hiku* (to pull back) and *komoru* (to withdraw) and brings attention to the main aspect of the phenomenon, namely, the withdrawal and isolation of an individual at home and from social relationships. The term can refer both to the individual living in social withdrawal and to the condition itself, a social behaviour that consists of a voluntary and prolonged detachment from physical and social realities of human experience. In 1978, Kasahara [4] reported clinical cases of *taikyaku shinkeishou* (withdrawal neurosis) suggesting the existence of a new condition that was characterized by social withdrawal. However, since then the term hikikomori has been used after the publication of the Japanese psychiatrist Tamaki Saito in 1998 [5]. Marked social isolation in one’s home, a duration of continuous social isolation of at least six months and significant functional impairment or distress associated with the social isolation are the diagnostic criteria for hikikomori according to the recent update by Kato, Kanba and Teo [6,7].

Hikikomori was initially presented as a typical phenomenon related to the Japanese culture, and Teo and Gaw [8] recommended additional research into whether hikikomori may be considered a psychological disorder more broadly. Subsequently, cases of hikikomori have been described worldwide, including Italy [2,9,10,11,12,13,14].

Epidemiological research carried out in several Asian countries has shown a prevalence of hikikomori between 1.2% and 2.3%, higher male vulnerability and onset in adolescence and young adulthood [15,16,17,18]. An online survey conducted in China estimated a prevalence of 6.6% for hikikomori with no apparent gender difference [19]. Recently, we found a frequency of 15.8% for lifetime hikikomori in a sample of young adults with clinical disorders and a frequency of 4.3% in a non-clinical sample recruited through an online survey in Italy [14].

Kato et al. [6] proposed a bio-psycho-socio-cultural model for the conceptualization of hikikomori, understood as a reaction to stress associated with psychological distress that might be separate from a clear diagnosis of psychiatric disorders. Personality and psychiatric disorders, such as avoidant, paranoid and obsessive compulsive personality disorders, depressive, psychotic, social anxiety and posttraumatic stress disorders may be comorbid with hikikomori [6,9,10,12,14,15,16,20]. Furthermore, overall personality dysfunction was associated with symptoms of hikikomori in clinical and non-clinical young adult populations [14]. Problems in family and social relationships, such as neglect, rejection, or bullying were also reported [6,13,16,21,22,23,24].

In addition to country-specific historical and cultural factors, some researchers and clinicians have focused their attention on the possible relationship between hikikomori and the worldwide spread of the use of the Internet and new technology [6,25,26]. In a study of 41 hikikomori in Korea, 56.3% were at elevated risk for Internet addiction while 9.4% were addicted [16]. Furthermore, around one third of 190 Spanish adults receiving help for social withdrawal at home showed symptoms of Internet addiction [12]. Finally, in a sample of Japanese emerging adults, participants at high risk for hikikomori had longer Internet usage time and higher problematic Internet and smartphone use than those not at risk [27].

Considering the expansion of this phenomenon, Teo et al. [28] developed the 25-item Hikikomori Questionnaire (HQ-25), a self-report scale evaluating the severity of symptoms of hikikomori, with a sample of Japanese participants from psychiatric and community settings. The authors first generated a list of 59 candidate items reflecting psychological and behavioral characteristics of hikikomori as reported by previous studies. Their analyses showed adequate psychometric properties and diagnostic accuracy of the HQ-25. Further, a three-factor solution emerged from the analysis, representing what the authors dubbed Socialization, Isolation and Emotional Support. Gundogmus et al. [29] performed an additional evaluation of the psychometric properties of the HQ-25 with a sample of healthy Turkish participants. Their findings supported the reliability and validity of the instrument. Hu et al. [30] reported encouraging findings on the psychometric properties of the Chinese version of the HQ-25. Furthermore, the questionnaire has been recently used in two studies [14,27].

Because of the public health and scientific concerns involving hikikomori in Italy, we believe a screening instrument to evaluate symptoms of hikikomori is needed to inform both clinical and research practices as well as to improve the dissemination of accurate information to the public, especially parents and teachers of individuals at risk for hikikomori. Thus, the research objectives of this study included: (1) to translate, adapt and validate the HQ-25, analyzing its psychometric properties for clinical and research use in Italian-speaking populations; and (2) to confirm the convergent validity of the Italian version of the HQ-25 exploring the relationship between symptoms of hikikomori and personality functioning, social support and problematic Internet use as well as comparing HQ-25 mean scores of participants reporting lifetime hikikomori with those of participants who do not report lifetime hikikomori.

## 2. Materials and Methods

### 2.1. Participants

A convenience sample of 410 participants with an age ranging from 18 to 73 years took part in this study. For analysis, all participants falling outside the age range of 18 to 50 years of age (*n* = 38) were excluded, similar to exclusion criteria used by Teo et al. [28] to ensure comparability between the age ranges of the samples. Therefore, the final sample size was 372 (73.1% females, *n* = 272) with a mean age of 31.55 (SD = 8.16). Regarding the geographical distribution of the participants, 23.9% lived in Northern Italy, 57.8% in the Center, 16.4% in Southern, and 1.9% in the Islands. Further, 80.9% (*n* = 301) had siblings, 73.4% (*n* = 273) reported a university or higher educational level and 12.9% (*n* = 48) lived alone.

### 2.2. Procedures

To limit response bias, this cross-sectional study was not specifically presented as a survey on episodes of social withdrawal and hikikomori but rather on mental health and the use of technology in general among Italian adults. The survey was conducted online using Google Forms. Thus, the informed-consent form and questionnaires were filled in online. The study used a non-probability sampling method. Survey participation was advertised using three social media platforms (i.e., WhatsApp, Facebook, and Twitter). Two classes of university students from the Faculty of Medicine and Psychology (Sapienza University of Rome) were encouraged to participate. Further, after completing the online questionnaires, participants were automatically invited to share the survey link with their personal contacts. Research data were encoded and stored on password protected drives with access limited to the research team. The study was approved by the Ethics Committee of the Department of Dynamic and Clinical Psychology, Sapienza University of Rome. The protocol for the research project conformed to the provisions of the Declaration of Helsinki.

### 2.3. Measures

#### 2.3.1. Hikikomori Symptoms

The 25-item Hikikomori Questionnaire (HQ-25) is a self-report questionnaire that explores the intensity of hikikomori symptoms over the preceding six months. Typical psychological features and behavioral patterns of hikikomori syndrome, such as socialization, isolation and emotional support, and a sense of alienation from society, are investigated. Participants respond on a 5-point Likert scale (from 0 = “strongly disagree” to 4 = “strongly agree”). The HQ-25 total score range is 0 to 100 with higher values indicating higher symptomatology. The authors showed that a cut off score of 42 was able to discriminate between individuals at risk for hikikomori and those not at risk with a sensitivity of 94% and specificity of 61% [28]. In the original study [28], Cronbach’s alpha coefficient for the whole scale was 0.96 (whereas for the three factors it was 0.94, 0.91 and 0.88, respectively). This result was confirmed by another study [14]. Convergent validity has been demonstrated with measures of loneliness, preference for solitude and (poor) social support [28] as well as interpersonal sensitivity and depression [14].

Standard forward and back translation procedures were used to ensure the equivalence of the meaning of questions in English and Italian. The questionnaire was translated into Italian by a native Italian speaker psychologist, and then translated back into English by a blinded translator who was not directly involved with the study. The two English-language versions were then compared showing that no further modifications were needed. The Italian version of the HQ-25 was reviewed by the first author (A.R.T.) of the original version of the questionnaire and is presented in Appendix B. In addition, we have subsequently adapted and validated the Italian version of the HQ-25 to be used with adolescents [31]. When this study was carried out there was no published literature on the Italian validation of the HQ-25; since then a different version of the HQ-25 with Italian participants has recently been used [32].

#### 2.3.2. Self-Reported History of Hikikomori

The definition of hikikomori proposed by Kato et al. [6,7] was adopted in this study. The following symptoms were examined for lifetime episodes of hikikomori through a set of questions designed for and used in a previous study [14]: (1) marked social isolation in one’s home (i.e., leaving the house 3 days/week or less); and (2) duration of social withdrawal of at least six months. Isolation caused by physical illness limiting the ability to walk or move, pregnancy or childbirth, working from home, or the need to stay at home to take care of children, were considered exclusion criteria of hikikomori, adopting a more stringent and conservative approach. Moreover, we examined the characteristics of individuals who exhibited social withdrawal behavior for at least three months but less than six (i.e., pre-hikikomori).

#### 2.3.3. Personality Functioning

Psychoticism subscale—Personality Inventory for DSM-5 Brief Form (PID-5BF) [33,34]. The PID-5-BF includes 25 items evaluating five maladaptive trait domains of personality functioning including psychoticism. Five items examined the psychoticism domain with four response choices distributed on a 4-point Likert scale (from 0 = “very false/often false” to 3 = “very true/often true”). Psychoticism increases as the score increases. The results obtained in the Italian validation study [34] showed that Cronbach’s alpha value for the psychoticism subscale was 0.77 and an adequate temporal stability at a two-month test–retest was found. Adequate internal consistency has been confirmed in a subsequent study [35]. Convergent validity with a measure of personality functioning was also demonstrated [34]. In the present study, Cronbach’s alpha was 0.71.

The Level of Personality Functioning Scale Brief Form 2.0 (LPFS-BF 2.0) [36] is a self-report instrument containing 12 items that provide a quick impression of personality functioning as described in Section III of the DSM-5. Participants respond on a 4-point Likert scale (from 1 = “completely untrue” to 4 = “completely true”). The LPFS-BF 2.0 has a score range of 12–48. Higher values indicate a higher severity of personality pathology. In the original study [36], the LPFS-BF 2.0 demonstrated satisfactory internal consistency (Cronbach’s alpha was 0.82) and promising construct validity in a sample of patients referred to a specialized center for personality disorders. Further, the scores were associated with related measures of psychopathology. A recent study confirmed that the instrument is able to capture impairment in personality functioning [37]. By fitting the model described in Weekers et al. [36] using data from our study, we obtained the following satisfactory fit indices from a Confirmatory Factor Analysis (CFA) of the LPFS-BF 2.0: Robust *χ^2^*(52) = 86.32, *p* = 0.002, RMSEA = 0.052, Robust RMSEA 90%C.I.: 0.031–0.070, SRMR = 0.047, Robust CFI = 0.951, Robust TLI = 0.938. Further, in the present study, Cronbach’s alpha was 0.84.

#### 2.3.4. Social Support

The Oslo Social Support Scale (OSSS-3) [38] was used to explore social support perceived by the participants through three items on the number of close confidants, sense of concern or interest from other people and relationships with neighbors [39]. Higher values indicate higher social support. Kocalevent et al. [39] obtained a Cronbach’s alpha of 0.64, considered satisfactory due to the brevity of the scale. Convergent validity with measures of psychopathology was also shown [40,41]. In the present study, Cronbach’s alpha was 0.50.

#### 2.3.5. Problematic Internet Use

The Internet Disorder Scale (IDS-15) Italian version [42,43] is a self-report scale composed of 15 items investigating the intensity and impact of problematic Internet use over the past year. Items are focused upon users’ online leisure activity from any device with Internet access. Respondents rate each item on a 5-point Likert scale (from 1 = “strongly disagree” to 5 = “strongly agree”). The IDS-15 has a total score range of 15–75, with higher scores being an indication of higher degrees of problematic Internet use. The Italian version of the IDS-15 showed excellent internal consistency (Cronbach’s alpha was 0.92) and participants’ scores correlated with hours of Internet use as well as measures of problematic videogame and social media use [42]. In the present study, Cronbach’s alpha was 0.88.

### 2.4. Statistical Analysis

Power analysis for determining the optimal sample size is reported in the Appendix A. Prior to proceeding to the confirmatory analysis, the normality of distribution for all 25 items of the HQ-25 was tested by checking whether skewness and kurtosis parameters were within the limits of ±2.00. To corroborate the three-factor structure of the questionnaire, we performed a CFA. Considering that not all items were normally distributed (at least 8 items showed a skewness or a kurtosis exceeding ± 2.00), we used the Unweighted Least Squares (ULS) estimator with robust standard error estimates (ULSM). The following fit indices were considered as measures of model fit: a non-significant robust chi squared statistics, the Comparative Fit Index (CFI) and the Bentler–Bonnett Non-Normed Fit Index (NNFI) (both above 0.90), the Root Mean Square Error of Approximation (RMSEA) and its 90% Confidence Interval (C.I.) (below 0.08) and the Standardized Root Mean Square Residual (SRMR) (below 0.08). All statistical analysis were performed using the R software [44] with the lavaan package [45], psych package [46] and coefficientalpha package [47]. Moreover, we estimated latent correlations among factors to verify the convergent validity of the HQ-25′s factors with the variables of interest (i.e., psychoticism domain and personality functioning, social support, problematic Internet use). Reliability was computed with Cronbach alpha and McDonald’s Omega [48].

Differences between groups (i.e., according to gender, HQ-25 cut-off, living condition, presence versus absence of siblings) were tested using student-t statistics and Cohen’s *d* as estimate of effect size. The examination of group differences based on the HQ-25 cut-off proposed in the original study by Teo et al. [28] was exploratory in nature considering that no analysis was performed to establish the most appropriate cut-off value for the Italian version of the HQ-25.

## 3. Results

The CFA demonstrated a satisfactory fit for the three-factor model to the data (Robust χ^2^(272) = 1232.6, *p* < 0.01, Robust CFI = 0.965, Robust NNFI = 0.962, Robust RMSEA = 0.089, 90%C.I. for Robust RMSEA: 0.084; 0.094; SRMR = 0.072). The one-factor model also showed a good fit (Robust χ^2^(275) = 1367.9, *p* < 0.01, Robust CFI = 0.960, Robust NNFI = 0.956, Robust RMSEA = 0.095, 90%C.I. for Robust RMSEA: 0.090; 0.100; SRMR = 0.077). Nevertheless, the fit of the three-factor model was significantly better than that of the one-factor model (Δχ^2^(3) = 17.6, *p* < 0.001).

The standardized factor loadings for the three-factor structure and latent correlations among the three factors are presented in Table 1. The three factors correlated positively (*p* < 0.001).

Cronbach alpha and omega, both demonstrated satisfactory reliability (socialization: Cronbach alpha = 0.90; 95%C.I.: 0.89; 0.92; Omega = 0.92, se = 0.009; isolation: Cronbach alpha = 0.83; 95%C.I.: 0.80; 0.85; Omega = 0.84, se = 0.017; emotional support: Cronbach alpha = 0.65; 95%C.I.: 0.59; 0.70; Omega = 0.65, se = 0.030).

### 3.1. Relationship between HQ-25 Factors and Other Psychological Constructs

Subsequently, we computed the latent correlations for testing whether the three HQ-25 factors correlated with the Psychoticism subscale score, Level of Personality Functioning Scale score, Internet Disorder Scale score and the Oslo Social Support Scale score. As shown in Table 2, all three HQ-25 factors correlated positively with psychoticism, level of personality functioning and problematic Internet use, and negatively with social support.

### 3.2. Mean Differences for the HQ-25′s Total and Three Factors’ Scores

Females and males showed no significant difference on the HQ-25 total score and subscores (Table 3).

### 3.3. Differences between Participants According to HQ-25 Cut-Off

Considering the cut-off score indicating “at risk of hikikomori” (HQ-25 score equal or above 42) we found 60 participants (16.2% of the sample; female *n* = 49; male *n* = 11) exceeding the threshold score. The “at risk” group was younger on average than the “not at risk” group (Table 4). We further investigated whether the two groups differed on the psychological construct scales. The “at risk” group for hikikomori showed significantly higher scores on the Psychoticism subscale, Level of Personality Functioning Scale and Internet Disorder Scale compared to the “not at risk” group, while they showed lower values on the Oslo Social Support Scale.

### 3.4. Differences among Participants Who Reported Lifetime Episodes of Hikikomori, Pre-Hikikomori and No History of Either

The HQ-25 total score and subscores for “lifetime hikikomori” and for “lifetime pre-hikikomori” compared with participants who did not report lifetime episodes of hikikomori are illustrated in Table 5. Four participants (among females: 1.1%, *n* = 3; among males: 1%, *n* = 1) reported lifetime episodes of hikikomori and five participants (among females: 1.8%, *n* = 5; among males: 0%) reported lifetime pre-hikikomori.

### 3.5. Sensitivity Analysis

Finally, we ran a series of comparisons between participants who reported living alone and those who did not live alone as well as between participants who had siblings and those who had not, with respect to their HQ-25 scores, showing no significant differences (Appendix A).

## 4. Discussion

Findings from the present study determined the reliability and validity of the Italian version of the HQ-25, thus encouraging its use as part of the clinical assessment of hikikomori. Initially, CFA demonstrated that a three-factor model consisting of Socialization, Isolation and Emotional Support showed a better fit than the one-factor solution. This provides confirmation of the three-factor structure found in the original study by Teo et al. [28]. Our study represents a first effort to provide data on the cross-cultural applicability of the HQ-25 with Italian adults suggesting its usefulness in capturing hikikomori symptoms. Second, we demonstrated that the instrument showed reliable criterion validity. The three factors obtained reflect behavioral and emotional aspects that correlated positively with personality dysfunction and problematic Internet use, and negatively with social support. Thus, the HQ-25 captures some core elements of the hikikomori phenomenon (i.e., difficulty in the socialization process, isolation and lack of emotional support) [6,14]. Moreover, some differences emerged between the three-factors of the HQ-25 according to the degree of association with other constructs of interest. The degree of association between emotional support and poor social support was higher than that observed between the other two factors (i.e., socialization and isolation) and poor social support. A similar pattern was observed for problematic Internet use and isolation. Despite the fact that other instruments exploring symptoms of hikikomori have been proposed [49,50], the HQ-25 has the added value of having been originally validated in both community participants and patients suffering from psychiatric conditions [28].

We observed a lower mean score (23.5 ± 16.8) on the HQ-25 than that (41.5 ± 22.3) reported by Teo et al. [28]. Our results are more in line with that (28.1 ± 16.3) observed by Tateno et al. [27], who recruited university students, than with that of Teo et al. [28]. Thus, these differences may be due to differences in sample recruitment and composition (e.g., community vs. clinical participants, age of participants). In line with Tateno et al. [27], we found no association between gender and HQ-25 scale scores. Approximately 16.2% of participants were at risk for hikikomori according to the HQ-25 cut-off proposed by Teo et al. [28]. Significant and positive associations emerged between symptoms of hikikomori, risk for hikikomori, and maladaptive personality traits. These findings are in line with those of previous studies [9,12,51] and are consistent with the hypothesis that hikikomori and its symptoms may represent the result of the dysfunctional adaptation process of individuals with high overall personality dysfunction in the context of increased social and environmental demands [14]. Further, we showed that severity of hikikomori symptoms were associated with poor social support and problematic Internet use confirming previous evidence [16,23,27]. However, a lack of association between symptoms of hikikomori and problematic Internet use have also been reported [9,14]. If problematic Internet use is a consequence of hikikomori, it may be the result of the need to maintain contact with reality or represent a way to deal with difficulties in social relationships (e.g., increasing social skills or finding social support online). On the other hand, it may also reflect a subsequent and higher degree of denial of and detachment from reality. If hikikomori is a consequence of problematic Internet use, however, problematic Internet use may lead to a gradual withdrawal from the real, physical world, exacerbating avoidance tendencies [26,52].

Prevalence of lifetime hikikomori and pre-hikikomori were 1.1% (*n* = 4) and 1.3% (*n* = 5), respectively. This result is in line with those of previous studies [15,16,17,18] and shows that social withdrawal and detachment from social realities is a phenomenon encountered in Italy. Taken together our findings have important clinical implications suggesting the usefulness of psychological interventions aimed at increasing the psychosocial skills of participants with or at risk of hikikomori.

Finally, we found no significant difference of HQ-25 scores between participants living alone and not-alone, and between participants who had siblings and those who had not. Accordingly, a case–control study [20] showed that relationship status such as being single or married was not associated with hikikomori. Taken together, these results suggest that simple indicators of social isolation may not be a risk factor for hikikomori, whereas more complex psychological traits do appear to be risk factors for hikikomori.

The present study is not exempt from limitations. First, the cross-sectional study design does not provide the opportunity to demonstrate causal inferences on the relationships between symptoms of hikikomori and the variables of interest. Second, data were collected using only self-report measures and, thus, susceptible to response biases. Third, the sample was composed of adults in the general population. Therefore, generalizations of our results to other groups or populations cannot be made. Even though all factor loadings were significant and congruent with the expected HQ-25 factors structure, two factor loadings were unexpectedly low (i.e., item 7: “There are people in my life who try to understand me”, and item 21: “I have someone I can trust with my problems” with factor loadings of 0.151 and 0.312, respectively, on the HQ-25 emotional support factor). Such factor loadings could be attributed to differences in sampling procedure between our validation study and Teo et al.’s [28] study. It is possible that in healthy adults, at least those included in this study, those items do not adequately represent emotional support as much as they do in clinical participants such as in the original study by Teo et al. [28]. Past validation studies of the HQ-25 did not find low factor loadings [29] or did not report information about factor loadings [30]. Such differences may also be due to “naturally expected” cultural differences between the Italian population and Japanese or Turkish populations. Our study was not aimed at studying the invariance of the Japanese and Italian factor structures (i.e., multi-sample CFA) and in absence of such kind of results we cannot conclude whether the factor loadings of the two items in the Italian version of the HQ-25 statistically differ from those of the Japanese version of the HQ-25. To solve this ambiguity, the invariance of the HQ-25 factorial structure across different cultures can be a valuable objective for future study, including not only Japanese and Italian populations but also Turkish and Chinese populations. Fourth, longitudinal studies are needed to clarify the causal nature of the associations between hikikomori and problematic Internet use. Finally, additional work to determine an optimal cut-off to improve clinical validity of the Italian version is needed.

## 5. Conclusions

The present study validated the Italian version of the HQ-25, providing additional evidence on the use of the HQ-25 as a reliable self-report questionnaire to investigate symptoms of hikikomori in psychological research and clinical practice.

## Figures and Tables

**Table 1 ijerph-19-13552-t001:** Completely standardized factor loadings and latent correlations among the three factors of the HQ-25.

	Socialization	Isolation	Emotional Support
Item 1	0.704		
Item 4	0.495		
Item 6	0.646		
Item 8	0.784		
Item 11	0.633		
Item 13	0.749		
Item 15	0.602		
Item 18	0.730		
Item 20	0.781		
Item 23	0.762		
Item 25	0.626		
Item 2		0.576	
Item 5		0.648	
Item 9		0.773	
Item 12		0.574	
Item 16		0.404	
Item 19		0.726	
Item 22		0.533	
Item 24		0.797	
Item 3			0.755
Item 7			0.151
Item 10			0.532
Item 14			0.548
Item 17			0.513
Item 21			0.312
Socialization	1		
Isolation	0.901	1	
Emotional Support	0.718	0.771	1

All parameters are significant at *p* < 0.01 with the exclusion of item 7 that was significant at *p* < 0.05. Between-factor correlations were significant at *p* < 0.001. *N* = 372.

**Table 2 ijerph-19-13552-t002:** Latent correlations among the three HQ-25 factors and variables of interest.

	Socialization	Isolation	Emotional Support
Psychoticism	0.554	0.576	0.595
Level of personality functioning	0.554	0.578	0.547
Social support	−0.377	−0.367	−0.641
Problematic Internet use	0.377	0.418	0.404

All latent correlations are significant at *p* < 0.01. *N* = 372.

**Table 3 ijerph-19-13552-t003:** Statistics for the HQ-25 according to gender.

	Total (*N* = 372)			Female (*n* = 272)			Male (*n* = 100)			*t*(370)	Cohen’s *d*	*p*
	M	SD	Min–Max	M	SD	Min–Max	M	SD	Min–Max			
Total HQ-25 score	23.50	16.78	0–81	23.30	16.72	0–81	24.03	17.01	0–76	−0.37	0.043	0.71
Socialization subscore	11.00	8.95	0–41	10.97	8.99	0–41	11.06	8.90	0–36	−0.08	0.009	0.93
Isolation subscore	7.27	6.06	0–31	7.27	6.09	0–31	7.28	6.00	0–27	−0.02	0.002	0.99
Emotional support subscore	5.23	4.09	0–24	5.06	4.06	0–24	5.69	4.14	0–15	−1.32	0.155	0.19

**Table 4 ijerph-19-13552-t004:** Statistics for scales exploring psychological symptoms of all participants, participants not at risk for hikikomori (HQ-25 score below the cut-off score) and participants at risk for hikikomori (HQ-25 score equal or above the cut-off score).

	Total (*N* = 372)			Not At Risk (*n* = 312)			At Risk (*n* = 60)			*t*(370)	Cohen’s *d*	*p*
	M	SD	Min–Max	M	SD	Min–Max	M	SD	Min–Max			
Age	31.55	8.16	18–50	32.00	8.20	18–50	29.10	7.70	19–50	2.55	0.360	0.011
Psychoticism	0.72	0.63	0–2.4	0.64	0.59	0–2.4	1.16	0.69	0–2.4	6.10	0.860	<0.001
Level of personality functioning	18.55	4.58	12–41	17.81	4.10	12–41	22.43	5.04	12–39	7.70	1.085	<0.001
Social support	10.12	1.94	3–14	10.39	1.86	3–14	8.73	1.80	4–13	5.85	0.825	<0.001
Problematic Internet use	34.48	10.59	15–68	33.13	9.98	15–68	41.50	11.00	18–65	6.37	0.898	<0.001

**Table 5 ijerph-19-13552-t005:** Descriptive statistics for the HQ-25, according to lifetime history of hikikomori.

	Hikikomori (*n* = 4)			Pre-Hikikomori (*n* = 5)			No Social Withdrawal (*n* = 363)		
	M	SD	Min–Max	M	SD	Min–Max	M	SD	Min–Max
Age	29.0	12.8	20–48	28.4	12.2	21–50	31.62	8.07	18–50
Total HQ-25 score	39.0	29.1	18–81	36.8	13.9	16–50	23.14	16.56	0–77
Socialization subscore	20.0	15.5	6–41	17.6	9.2	8–28	10.81	8.82	0–38
Isolation subscore	12.0	12.2	4–30	12.4	5.7	5–20	7.15	5.95	0–31
Emotional support subscore	7.0	2.2	5–10	6.8	4.2	3–14	5.19	3.26	0–24

## Data Availability

The data presented in this study are available on request from the corresponding author.

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
