# Peer review of "Psychometric Properties of the Italian Version of the 25-Item Hikikomori Questionnaire"

_ijerph, 2022, doi:10.3390/ijerph192013552_

Round 1

Reviewer 1 Report

The article presents an interesting study. The topic dealt with is very original, the in-depth study of the literature and the validation study appear accurate, the text is written fluently and clearly. However, there are some aspects that should be improved.

At first, from the introduction it would seem that the phenomenon concerns mainly young adults. I guess that it was for this reason that you excluded the over 50 from the analysis. Is it correct? In the paragraph "participants", you only say "similar to exclusion criteria used by teo et al", but it would be better to explain the ratios behind the exclusion.

Moreover there is a small discordance: first you say that the participants were a minimum of 18 years and then that you excluded those under 15. So were participants under 18 (and so minors) or not?

It would be helpful to have more information on the sample. Did the participants all live in Italy? If you have also collected information about the territorial distribution (north, center, or south of Italy) of the participants, it would be useful to make it explicit. If, on the other hand, the sample was collected only in a certain area of the country, the "local" nature of the sample should be reported as a limitation of the research.

Information on the occupational status of the sample would also be useful, to understand if there is a prevalence of a professional category.

Why did you include "to have siblings" in the demographic variables? Is there any particular relevance in relation to the hikikomori phenomenon? If so, it should be discussed.

You write "Informed consent was conducted online". Did you mean that the questionnaire (and also the consent request) was submitted online? If so, through which platform?

Furthermore, the authors should explain how they disseminated the questionnaire (and I imagine an access link). What was the sampling technique used? Was the research advertised through university students, an agency, or personal contacts?

Finally, the biggest issue of the study concerns the poor factor loadings for items 7 and 21 (item 7 is not even significant at the .01 level). At the moment you explain it by saying "is possible that in healthy adults, at least those included in this study, those items do not adequately represent emotional support as much as they do in clinical participants such as in the original study by Teo et al". However, reading the items, this explanation does not seem convincing, to explain why these two items do not work well.

I believe the authors should discuss this point further. Did this problem arise in other validation studies as well? And in the other cases how was it explained?

Did the authors also try to test a model that considered possible correlations between the errors of these items or excluded them?

Reviewer 2 Report

The authors present a paper on the psychometric properties of the Italian sample engaging in prolonged and extreme social withdrawal lasting at least six months. That is commonly termed hikikomori, from the Japanese word referring to a condition of self-confinement and withdrawal. Even though there are other scales as Hikikomori Behavior Checklist or Hikikomori Risk Scale however in this paper the authors use an instrument to date is based on a widely accepted theoretical definition of hikikomori the Hikikomori Questionnaire (HQ-25)

A great deal of work has been published on the use of this questionnaire in different European countries. This paper focuses on Italy and is an extension of a previous work carried out by the same authors with Italian adolescents.

The first question I would like to ask the authors is why a new study on the psychometric properties of the Hikikomori Questionnaire has been published on the Italian validation of the Hikikomori Questionnaire (HQ-25). These two very similar articles however differ in something important the authors use a sample of people who do not present mental problems while an article with the same purpose has been carried out in 2021 by Emanuele Fino and his colleagues. (Fino, E., Iliceto, P.,Carcione,A.,Giovani, E., & Candilera, G. (2022) Validation of the Italian version of the 25-Item Hikikomori Questionnaire (HQ-25), Journal of Clinical Psychology, 1-18. https://doi.org/10-1002/jclp.23404

Fino et al. (2022) do use the strategy of Teo et al. (2018) with two samples, one the subsample of healthy subjects and another of psychiatric hospital patient subjects. However, Amendola et al. (2022) and in this article present a single convenience sample without health problems.

The fact that it does not appear in the bibliography is significant. It could also be said that their work does not bring originality, since they were already published in this same journal, by the same authors, the Italian version for adolescents. It is true that the factors for validating the scale are different, but this does not obviate the fact that it is an extension of the research you have already carried out.

Statistically, the tests are the correct ones doing a confirmatory factor analysis to corroborate the three-factor model of Teo et al. (2018), the second question I would like you to clarify is why you did not use the same factors as in the "adolescents" article. That is, Psychoticism subscale - Personality Inventory for DSM-5 Brief Form (PID-5BF); Internet Disorder Scale (IDS- 15); the Brief Symptom Inventory (BSI), and the Brief Prodromal Questionnaire (PQ-B).

However, in this article, the Oslo Social Support Scale (OSSS-3) and the level of personality functioning scale (LPFS-BF 2.0) are incorporated and the Brief Symptom Inventory (BSI) and the Brief Prodromal Questionnaire (PQ-B) are removed. This should be clarified in order to understand the reason for your research.

Also, it presents a problem in their article on the HQ-25 in Italian adolescents they use an age range that goes from 11 to 19 years, instead in the adult sample it goes from 15 to 50 just like Teo et al. (2018). You should explain this issue because there would be four years shared between adults and adolescents, i.e. the results would be debatable when using two ways of measuring age.

The Teo et al (2018) paper introduces 6 items that should be reversed (4,7,10,15,21, and 25 ) however the authors show item 22 as to be reversed scored, could you clarify why if in the English version scale (Teo et al. 2018) it is not reversed scored.

Round 2

Reviewer 1 Report

The authors responded effectively to all my observations.

In my opinion, the article is now improved and almost ready for publication.

I have just one last notation:

The authors have rightly added a further explanation (cultural differences) to motivate the low factor loadings of items 7 and 21. But this does not explain why only these two items are problematic.

Surely their proposal for future studies that investigate cultural differences between different countries is very valid, but it would be necessary first of all to hypothesize a further study in Italy, to deepen and solve (perhaps by modifying the translation) the issue posed by these two items.

I advise authors to further specify this aspect in their discussions.
